# The Impact of Inter-Basin Water Transfer Schemes on Hydropower Generation in the Upper Reaches of the Yangtze River during Extreme Drought Years

**Fan Wen** [1], **Mingxiang Yang** [1,2,*], **Wenhai Guan** [3], **Jixue Cao** [3], **Yibo Zou** [3], **Xuan Liu** [1,4], **Hejia Wang** [1] and **Ningpeng Dong** [1,2,5,6,*]

1    State Key Laboratory of Simulation and Regulation of Water Cycle in River Basin, China Institute of Water Resources and Hydropower Research, Beijing 100038, China; wenfan@edu.iwhr.com (F.W.); xuan_liu@tju.edu.cn (X.L.); hjwang@iwhr.com (H.W.)
2    Cooperative Innovation Center for Water Safety & Hydro Science, Hohai University, Nanjing 210000, China
3    China Three Gorges Corporation, Yichang 443000, China; guan_wenhai@ctg.com (W.G.); cao_jixue@ctg.com (J.C.); zou_yibo@ctg.com (Y.Z.)
4    College of Civil Engineering, Tianjin University, Tianjin 300354, China
5    State Key Laboratory of Hydrology-Water Resources and Hydraulic Engineering, Hohai University, Nanjing 210098, China
6    Key Laboratory of Flood and Drought Hazard Control, Ministry of Water Resources, Nanjing 210029, China
*    Correspondence: yangmx@iwhr.com (M.Y.); dongnp@iwhr.com (N.D.)

**Abstract:** The Yangtze River Basin experiences frequent extreme heatwaves and prolonged droughts, resulting in a tight supply demand balance of electricity and negatively impacting socioeconomic production. Meanwhile, ongoing inter-basin water diversion projects are planned that will divert approximately 25.263 billion cubic meters of water from the Yangtze River Basin annually, which may further affect the power supply in the region. In this study, the CLHMS-LSTM model, a land-surface hydrological model coupled with a long short-term memory (LSTM)-based reservoir operation simulation model, is used to investigate the impact of water diversions on the power generation of the Yangtze River mainstream reservoirs under extreme drought conditions. Two different water diversion schemes are adopted in this study, namely the minimum water deficit scheme (Scheme 1) and minimum construction cost scheme (Scheme 2). The results show that the land surface–hydrological model was able to well characterize the hydrological characteristics of the Yangtze River mainstem, with a daily scale determination coefficient greater than 0.85. The LSTM reservoir operation simulation model was able to simulate the reservoir releases well, with the determination coefficient greater than 0.93. The operation of the water diversion projects will result in a reduction in the power generation of the Yangtze River mainstem by 14.97 billion kilowatt-hours. As compared to the minimum construction cost scheme (Scheme 2), the minimum water deficit scheme (Scheme 1) reduces the loss of power generation by 1.38 billion kilowatt-hours. The research results provide new ideas for the decision-making process for the inter-basin water diversion project and the formulation of water diversion plans, which has implications for ensuring the security of the power supply in the water diversion area.

**Keywords:** LSTM; land-surface hydrological model; water diversion; power generation

## 1. Introduction

In arid regions, freshwater resources are becoming increasingly scarce [1]. As a result, rapid population growth and the expansion of industry and agriculture have led to a sharp increase in human demand for water resources [2]. In the context of global warming, frequent extreme drought events exacerbate conflicts between water supply and demand, creating increasingly greater challenges for water resource management and competition between different water-using sectors [3,4]. This leads to increased risks in water supply

and a growing contradiction between the increasing demand for water and the social development of water-deficient areas [5].

Implementation of inter-basin water transfer projects is an effective engineering measure to enhance water resource security through artificial water transfer, alleviate water scarcity caused by uneven distribution of water resources, and balance the distribution of water resources between basins [5,6]. To alleviate the potentially worsening water and energy scarcity problem by 2030, decision makers around the world are actively building dams and other types of water infrastructure [7,8]. In China, large-scale inter-basin water transfer projects have been planned and implemented to transfer water resources from water-rich areas to water-deficient areas [9–12]. However, inter-basin water transfer projects not only play an important supportive and protective role in promoting economic and social development in the receiving areas but also change the hydrological runoff situation of the supplying areas [13]. Therefore, it is particularly important to evaluate the potential impacts of inter-basin water transfer projects [14].

When transferring water from the supply area to the receiving area through inter-basin water transfer projects, the allocation of water resources unavoidably affects the water facilities and resource utilization in the supply area [15]. Among them, hydropower generation is an important way of comprehensively utilizing water resources and is a critical component of water conservancy projects [16]. However, the hydropower generation and the economic benefits of the reservoir are often impacted to varying degrees by the decreased water resources due to the inter-basin water transfer projects [17]. This is especially the case for the upstream of the Yangtze River in China, where the impact of inter-basin water transfer projects on hydropower generation can be more evident. The upstream of the Yangtze River is one of the largest hydropower development zones in China and an important energy base in western China [18]. Here, hydropower generation is a vital economic source. However, due to water loss caused by inter-basin water transfer projects, the hydropower generation in the supply area is often impacted to varying degrees [19]. Therefore, studying the impact of inter-basin water transfer projects on hydropower generation in the supply area of the upstream of the Yangtze River in China is of great theoretical and practical significance [20,21].

In addition, under extreme weather conditions, the changes in hydropower generation have a significant impact on the power system [22]. Currently, the frequency and intensity of extreme weather events are constantly increasing, which imposes higher requirements regarding the safety and stability of the power system [23]. Currently, numerous studies have been performed assessing the impacts of inter-basin water transfer projects on various aspects. These studies have examined the effects of inter-basin water transfer on water flow conditions in the receiving areas [24], the impact of climate change on the scale of inter-basin water transfer projects [25], as well as the ecological and human health issues arising from inter-basin water transfers [26]. In addition, these studies have also evaluated the effects of the operation of inter-basin water transfer projects on emergency water supply [27] and biodiversity [13,28] in the receiving areas. However, research on the quantitative impact of inter-basin water transfer projects on hydropower generation in the supply area is limited, especially for the upstream of the Yangtze River in China. Therefore, under the current background of climate change, studying the impact of inter-basin water transfer projects on hydropower generation in the supply area and the impact of extreme weather events on hydroelectric power systems is of great practical significance for promoting the construction of a new power system and ensuring energy security.

In this study, we aim to quantify and evaluate the impacts of inter-basin water transfer projects on the electricity generation and structure of the supply area under extreme drought conditions. To achieve this, we use a coupled land-hydrological model system and a long short-term memory (LSTM)-based reservoir simulation model to calculate the release and power generation of the reservoir [29]. At the same time, two water transfer schemes are taken into consideration. Water transfer Scheme 1 involves adjusting the water transfer amount on a monthly basis according to the design plan and parameters in the initial

report, in order to minimize water deficit. Water transfer Scheme 2 involves distributing the annual water transfer amount evenly across each month, under the premise of a constant total water transfer amount, in order to achieve a balanced and stable water transfer, and minimize construction costs.

Finally, we compare and analyze the impacts of the two water transfer schemes on electricity generation and structure in the supply area. Through quantitative analysis, we found that compared to water transfer Scheme 1, water transfer Scheme 2 can reduce the impact of inter-basin water transfer on the power generation of the cascade reservoirs of the Yangtze River mainstream, thus reducing the risk of insufficient power supply during the dry season. With the growth of the economy and population, the contradiction between water supply and demand will become increasingly prominent. Inter-basin water transfer projects, as an effective means of allocating water resources, will play an important role in future water resources management. This study provides a feasibility analysis of inter-basin water transfer schemes based on the generation and structure of electricity in cascade hydropower stations, which can serve as a reference and as guidance for other similar inter-basin water transfer projects. Additionally, the conclusions of this study can also serve as a reference for future management and allocation of water resources in the Yangtze River Basin, helping decision makers to formulate relevant policies and plans. Furthermore, with continuous technological progress and innovation, the construction and operation of inter-basin water transfer projects will become more intelligent and efficient, such as combining Internet of Things technology, artificial intelligence technology, and big data analysis to improve the efficiency of water resource utilization and reduce environmental impacts. Therefore, future inter-basin water transfer projects will not only be a simple water resource allocation, but also a comprehensive project involving technology, economy, environment, and other aspects, making greater contributions to sustainable development of humanity.

The structure of the remaining parts of this article is as follows: Section 2 introduces the data and schemes used in this study, Section 3 validates the models used, Section 4 provides a detailed description of the impacts of the water transfer project on the electricity generation of the cascade reservoirs in the upper reaches of the Yangtze River, and compares and analyzes the impacts of the two water transfer schemes on electricity generation and structure. Section 5 discusses the results. Finally, Section 6 presents the conclusion.

## 2. Materials and Methods

### 2.1. Study Area

In this paper, we selected the upper Yangtze River basin as the study area, which is located in southwest China (as shown in Figure 1), spanning from 90–111° E and 24–36° N, as the study area [30]. The Yangtze River runs from the headwaters in the Tibetan Plateau to the Yichang City, with a total length of approximately 4504 km and a drainage area of about 1 million $km^2$, accounting for 56% of the entire Yangtze River basin. The major cascade reservoirs in the upper Yangtze River basin include the Wudongde (WDD), Baihetan (BHT), Xiluodu (XLD), Xiangjiaba (XJB), Three Gorges (SX), and Gezhouba (GZB) reservoirs (as shown in Figure 1). Table 1 lists the characteristic parameters of the cascade reservoirs in the upper Yangtze River basin.

In the upper reaches of the Yangtze River basin, four inter-basin water transfer projects are under construction, namely the Hanjiang River diversion project (P1), the Central Yunnan Water Transfer project (P2), the Bailong River Water Diversion project (P3), and Phase I of the Western Route of the South-to-North Water Transfer project (P4), as shown in Figure 1. The Hanjiang River diversion project transfers water from the upper reaches of the Yangtze River to the water-receiving areas of the Central Route of the South-to-North Water Transfer project, the middle and lower reaches of the Han River in Hubei Province, the water-receiving areas of the Han River-Water Conservancy Project, and the supplementary water area on the right bank of the Han River in Hubei Province, with an estimated annual average water transfer of 3.9 billion cubic meters. The Central Yunnan Water Transfer

project transfers water from the Jinsha River to Xinpo Township in Honghezhou, with an estimated annual average water transfer of 3.403 billion cubic meters. The Bailong River Water Diversion project transfers water from the Daigusi Reservoir to the southeastern area of Longdong in Gansu Province, with an estimated annual average water transfer of 0.96 billion cubic meters. Phase I of the Western Route of the South-to-North Water Transfer project transfers water from the upstream of the Yalong River and Dadu River to provinces along the Yellow River, with an estimated annual average water transfer of 8 billion cubic meters. The data were extracted from the preliminary design reports of various water diversion projects and official public statements, ensuring their reliability. The specific information and estimated annual average water transfer of these inter-basin water transfer projects are presented in Table 2.

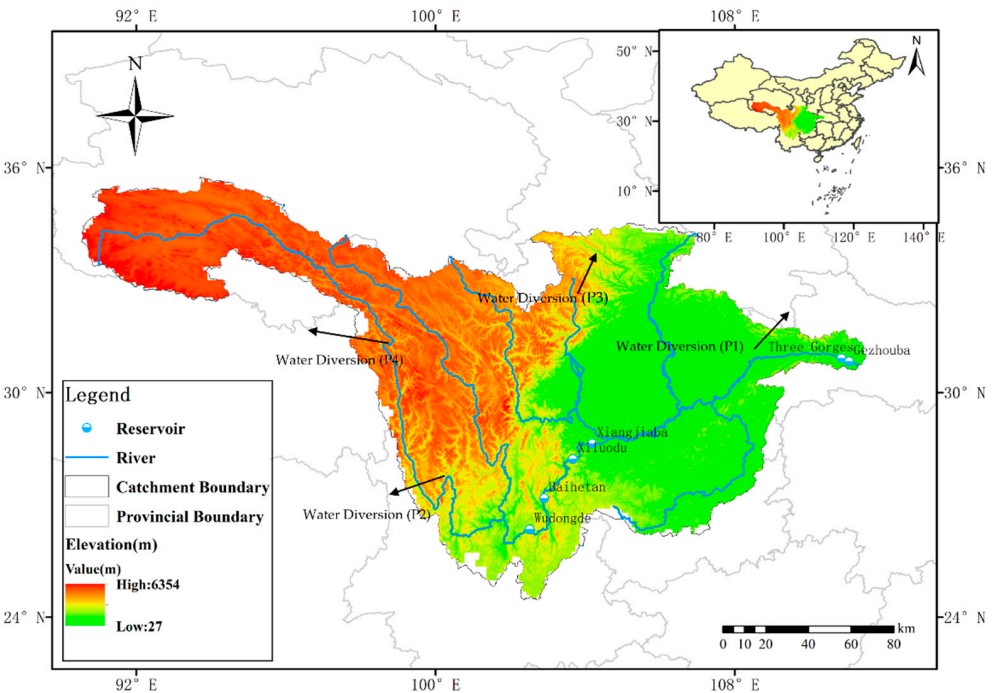

**Figure 1.** The upper reaches of the Yangtze River.

**Table 1.** Engineering characteristic parameters of cascade reservoirs in the upper reaches of the Yangtze River.

| Reservoir | WDD | BHT | XLD | XJB | SX | GZB |
|---|---|---|---|---|---|---|
| Normal operating level (m) | 975 | 825 | 600 | 380 | 175 | 66 |
| Flood control level (m) | 952 | 785 | 560 | 370 | 145 | 62 |
| Inactive level (m) | 945 | 765 | 540 | 370 | 145 | 62 |
| Flood control storage ($10^8$ m$^3$) | 24.4 | 75 | 46.5 | 9.03 | 221.5 | 7.11 |
| Conservation storage ($10^8$ m$^3$) | 32.2 | 104.36 | 64.6 | 9.03 | 221.5 | 7.11 |
| Installed capacity (MW) | 10,200 | 16,000 | 13,860 | 6400 | 22,500 | 2735 |

**Table 2.** Description of the water transfer projects in the upper reaches of the Yangtze River.

| Project Name | Water Source | Water Receiving Area | Annual Water Diversion Amount (Billion m$^3$) |
|---|---|---|---|
| P1 | Upstream of the Three Gorges Dam | Hubei Provinc | 3.9 |
| P2 | Jinsha River | Honghezhou | 3.403 |
| P3 | Daikosi Reservoir | Gansu Province | 0.96 |
| P4 | Shuangjiangkou | yellow river basin | 8.0 |

### 2.2. CLHMS Model

In this study, the land surface–hydrological model CLHMS, which is a fully coupled system of the land surface scheme LSX and the distributed hydrological model HMS, is adopted to simulate the hydrological regime [29]. The LSX model consists of modules for soil, vegetation, snow, glacier, etc. Based on the water-heat balance, it calculates the runoff, evapotranspiration, and water flux at the bottom of soil using meteorological variables such as precipitation, solar radiation, air temperature, wind speed, air pressure, specific humidity, cloud cover, etc., and transfers them to the distributed hydrological model HMS. The HMS model includes one-dimensional river routing module, two-dimensional hillslope routing module, two-dimensional lake hydrodynamics module, two-dimensional groundwater hydrodynamics module, etc., and can simulate surface water and groundwater routing, which calculates the soil moisture content and groundwater level in the deep vadose zone and feedbacks them to the LSX model to update the water flux at the bottom of soil, forming a complete land surface–hydrological two-way coupling model.

In this study, daily meteorological data required to drive the coupled land surface–hydrological model include the CN05.1 precipitation dataset (http://data.cma.cn, accessed on 15 April 2022) and the NCEP/NCAR reanalysis data (http://rda.ucar.edu/datasets, accessed on 15 April 2022). The model runs at a daily scale with a spatial resolution of 5 km.

### 2.3. LSTM-Based Reservoir Operation Simulation Model

In this study, we developed a Long Short-Term Memory (LSTM)-based reservoir operation model to simulate the reservoir storage and releases in a similar manner to Dong et al. [29]. LSTM is a variant of Recurrent Neural Network (RNN) specifically designed for processing sequential data and consists of a set of memory cells that control the flow of information through gate mechanisms [31]. These gates regulate the input, output, and forgetting of information to process long-term dependencies [32]. Although LSTM has a relatively complex structure and requires a large amount of training data and computational resources, it has the advantage of handling long sequence data and has broad applications in the field of hydrology and water resources [33]. However, like any other machine learning model, LSTM models also have their limitations and drawbacks such, including overfitting and high computational cos. The reliability of the model can be demonstrated through model validation, and the computational cost can be reduced by optimizing the computational process.

The process of developing LSTM network includes the following steps: First, the identification of the necessary input information and passing it to the input gate of LSTM, which decides whether to allow the information to pass based on the output value of the sigmoid function [34]. Secondly, the use of another sigmoid function to determine which information should be retained in the LSTM cell. This gate is called the output gate, which determines what information should be output. Finally, a tanh function is used to generate a new cell state and combining it with the previous cell state to generate a new output. By following these steps, an LSTM network with memory capability can be constructed, which can effectively process time-series data [35]. The structure of the LSTM neural network is shown in Figure 2.

The data required to train the LSTM model in this study included historical inflow, water level, and outflow data for the Xiluodu, Xiangjiaba, and Three Gorges reservoirs. Due to the short operating history of the Wudongde and Baihetan reservoirs, there are no long series of historical operation data. Therefore, the operation mode of the Wudongde and Baihetan reservoirs adopts a conventional reservoir scheduling model based on the reservoir scheduling chart according to their respective operating rules. The Gezhouba reservoir is treated as a run-of-river reservoir due to its negligible regulation capacity.

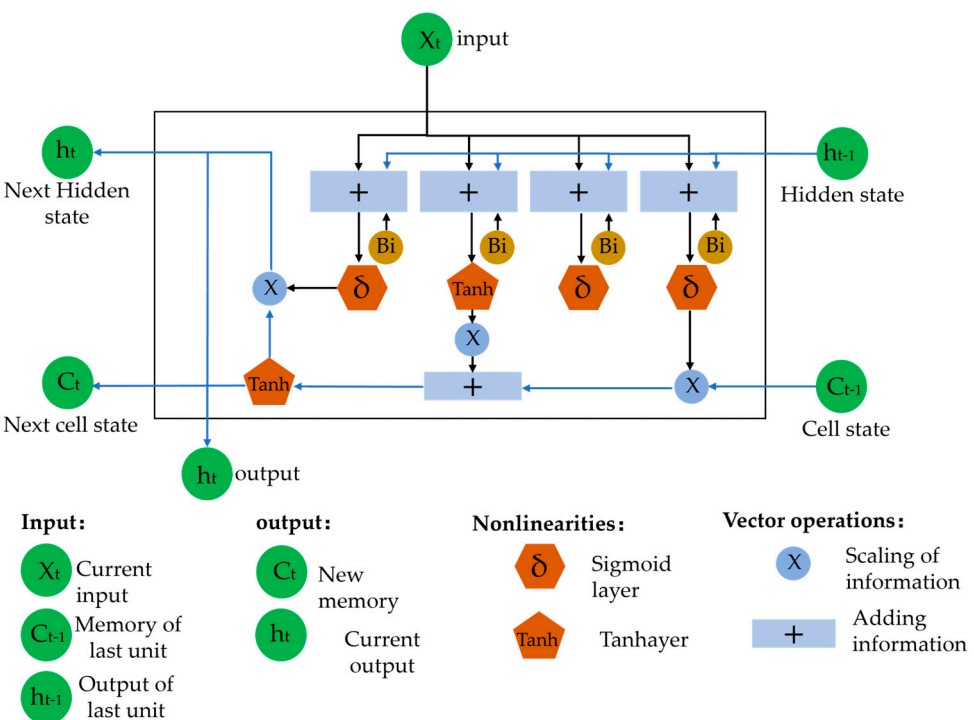

**Figure 2.** The structure of the LSTM neural network.

### 2.4. Experimental Design

In this study, the coupled land surface–hydrological model (CLHMS) is used to simulate the daily average inflow data for the Wudongde reservoir from 1981 to 2010, as well as the daily average interval inflow data for five other reservoirs including Baihetan, Xiluodu, Three Gorges, Gezhouba, and Xiangjiaba during the same period. The P-III curve is used to fit the daily flow data for the extremely dry year to obtain the daily inflow data for the Wudongde reservoir and the interval inflow data for each reservoir. The hydrological data used in this study are from China's official hydrological yearbooks, which are considered reliable.

This study used the coupled land surface–hydrological model (CLHMS) to simulate the daily average inflow data of Wudongde Reservoir from 1981 to 2010, as well as the daily average interval inflow data of five reservoirs (Baihetan, Xiluodu, Xiangjiaba, Three Gorges, and Gezhouba) from 1981 to 2010. The P-III curve is used to fit the daily flow data for extreme drought years, providing the daily inflow data for Wudongde Reservoir and the inflow data for each reservoir interval.

The aim of this study is to analyze the impact of water diversion projects on reservoir hydropower generation. To achieve this goal, we adopt two different water diversion schemes in this study, namely the minimum water deficit scheme (Scheme 1) and the minimum construction cost scheme (Scheme 2). Scheme 1 determines the water diversion amount on a monthly basis according to the net water demand from basin-level design reports, in order to minimize the water deficit and achieve optimal water diversion effects. Scheme 2 distributes the annual water diversion amount evenly across the months to achieve a balanced and stable water diversion process. Since the construction cost of diversion pipeworks, and hence the water diversion project, is related to the maximum monthly water diversion amount, Scheme 2 can reduce the maximum monthly water diversion amount and hence minimize the construction cost of the project.

Furthermore, the formulation of new water diversion schemes is expected to alleviate the impact of water diversion on the hydropower generation of the Yangtze River cascade, since the water diversion scheme for the South-to-North Water Diversion West Route Project and the Han River Diversion Project has not yet been fully determined. Therefore,

the water diversion schemes proposed in this study could provide valuable references for the design and practice of water diversion projects. The specific water diversion schemes are detailed in Table 3.

**Table 3.** Water diversion volume for each month in the water diversion project.

| Project Name | Water Diversion Schemes | Water Transfer Scale by Month in Extremely Dry Year (m³/s) | | | | | | | | | | | | Total |
|---|---|---|---|---|---|---|---|---|---|---|---|---|---|---|
| | | 1 | 2 | 3 | 4 | 5 | 6 | 7 | 8 | 9 | 10 | 11 | 12 | |
| P1 | scheme 1 | 99 | 87 | 76 | 104 | 208 | 211 | 171 | 165 | 117 | 105 | 117 | 102 | 1562 |
| | scheme 2 | 130 | 130 | 130 | 130 | 130 | 130 | 130 | 130 | 130 | 130 | 131 | 131 | 1562 |
| P2 | scheme 1 | 16 | 16 | 17 | 15 | 21 | 15 | 22 | 24 | 20 | 15 | 16 | 16 | 215 |
| | scheme 2 | 18 | 18 | 18 | 18 | 18 | 18 | 18 | 18 | 18 | 18 | 18 | 17 | 215 |
| P3 | scheme 1 | 0 | 0 | 0 | 14 | 40 | 52 | 61 | 41 | 35 | 30 | 23 | 4 | 299 |
| | scheme 2 | 25 | 25 | 25 | 25 | 25 | 25 | 25 | 25 | 25 | 25 | 25 | 24 | 299 |
| P4 | scheme 1 | 115 | 120 | 130 | 190 | 329 | 369 | 315 | 282 | 171 | 118 | 112 | 110 | 2359 |
| | scheme 2 | 197 | 197 | 197 | 196 | 196 | 196 | 196 | 196 | 197 | 197 | 197 | 197 | 2359 |

## 3. Model Calibration and Validation

### 3.1. Validation and Calibration of CLHMS Models

To verify the feasibility of the coupled land surface–hydrological model, we conducted calibration and validation of the runoff parameters at the Yichang, Cuntan, and Pingshan stations. We selected the period from 1981 to 1994 as the calibration period and the period from 1995 to 2020 as the validation period. We simulated the maximum average daily Nash–Sutcliffe efficiency (NSE) values of the model's randomly generated 10,000 parameter sets and selected the best performing parameter set as the optimal one.

Figure 3 shows the daily observed and simulated runoff at the three hydrological stations. Our model performed well at these three stations, with a daily deterministic coefficient greater than 0.9 during the calibration period and greater than 0.85 during the validation period. This indicates that our hydrological model can simulate the hydrological characteristics of the Yangtze River mainstream well.

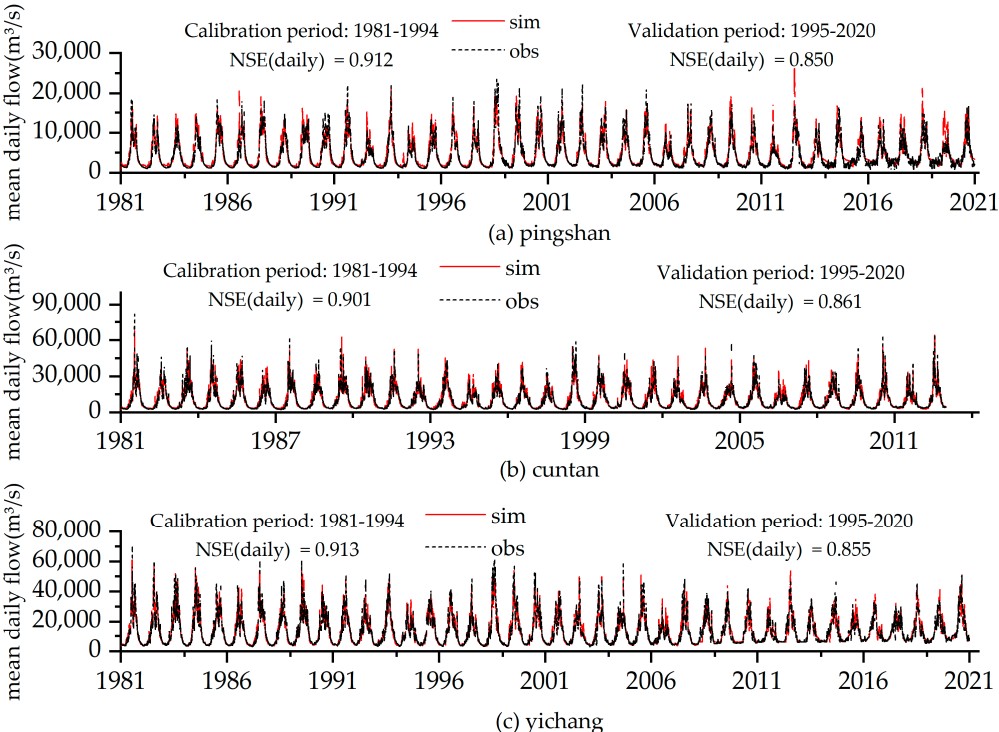

**Figure 3.** The simulated and observed daily streamflow at (**a**) Pingshan, (**b**) Cuntan, and (**c**)Yichang.

### 3.2. Validation and Calibration of LSTM-Based Reservoir Operation Simulation Model Models

To verify the applicability of the LSTM model in reservoir operation simulation, we calibrated and validated the model using historical inflow, water level, and outflow data from the Xiluodu, Xiangjiaba, and Three Gorges reservoirs. Specifically, we selected the period from 2014 to 2018 as the calibration period for Xiluodu Reservoir and the period from 2019 to 2021 as the validation period. For Xiangjiaba Reservoir, we chose the period from 2013 to 2018 as the calibration period and the period from 2019 to 2021 as the validation period. For the Three Gorges Reservoir, we selected the period from 2009 to 2016 as the calibration period and the period from 2017 to 2021 as the validation period.

Figure 4 shows the observed and simulated outflow of Xiluodu, Xiangjiaba, and Three Gorges reservoirs. Our reservoir model performed well on these three reservoirs, with daily coefficient of determination (R-squared) values greater than 0.94 in the calibration period and greater than 0.93 in the validation period. This indicates that the LSTM reservoir simulation model can simulate the storage and release processes of reservoirs well, demonstrating its applicability.

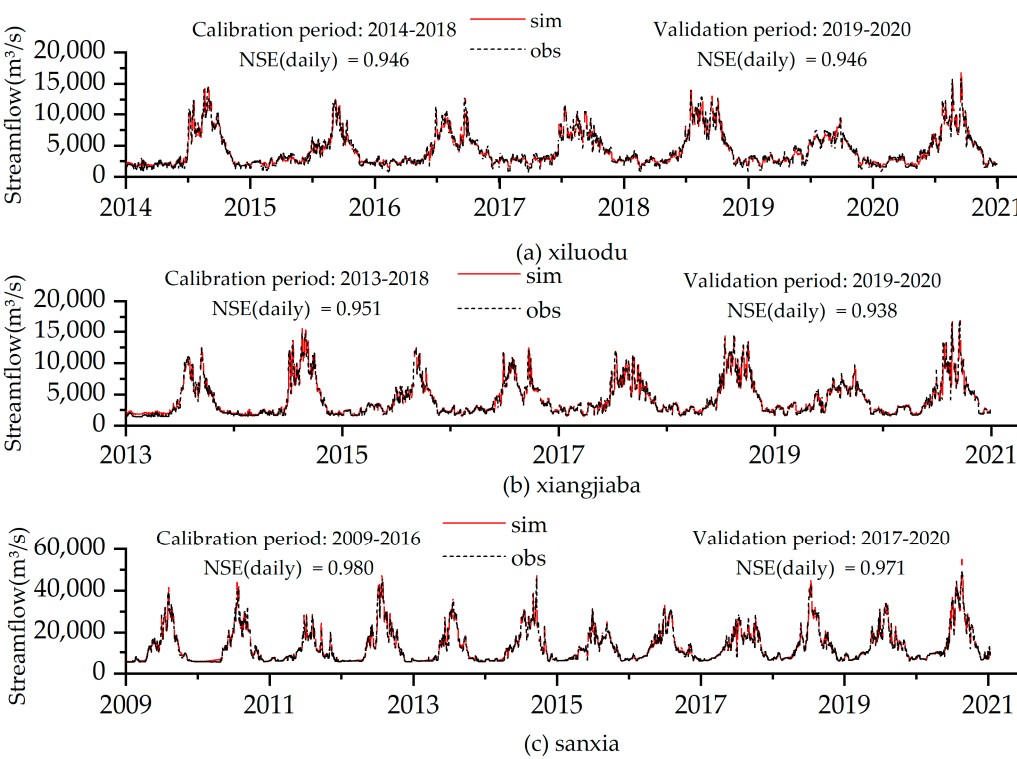

**Figure 4.** Comparison of observed and simulated daily runoff from reservoir outflows.

## 4. Impacts of Reservoir Operation

This study focuses on the cascade reservoirs of the Yangtze River as the research object. Combining two water diversion schemes and using the CLHMS-LSTM model developed in this paper, the impact of inter-basin water diversion on the power generation of the Yangtze River cascade reservoirs is analyzed.

### 4.1. Impacts of Water Transfer Scheme 1 on Reservoir Generation

Based on the water inflow data of each reservoir under extreme drought conditions and the water transfer data of each water diversion project, we used the CLHMS-LSTM model to solve and calculate the water levels and power generation of each cascade reservoir in the upper reaches of the Yangtze River.

Figure 5 shows the monthly power generation of the cascade power stations under the water transfer scheme. The results indicate that the inter-basin water transfer has an impact

on the power generation of the reservoirs in each month. Among them, the Wudongde, Baihetan, Xiluodu, Xiangjiaba, Three Gorges, and Gezhouba reservoirs experience the most significant power generation losses in May, with the Three Gorges and Gezhouba reservoirs experiencing the largest power generation losses. After water transfer, the power generation of each reservoir in each month is generally lower than before the water transfer, but the power generation of the Gezhouba reservoir in October is higher after water transfer.

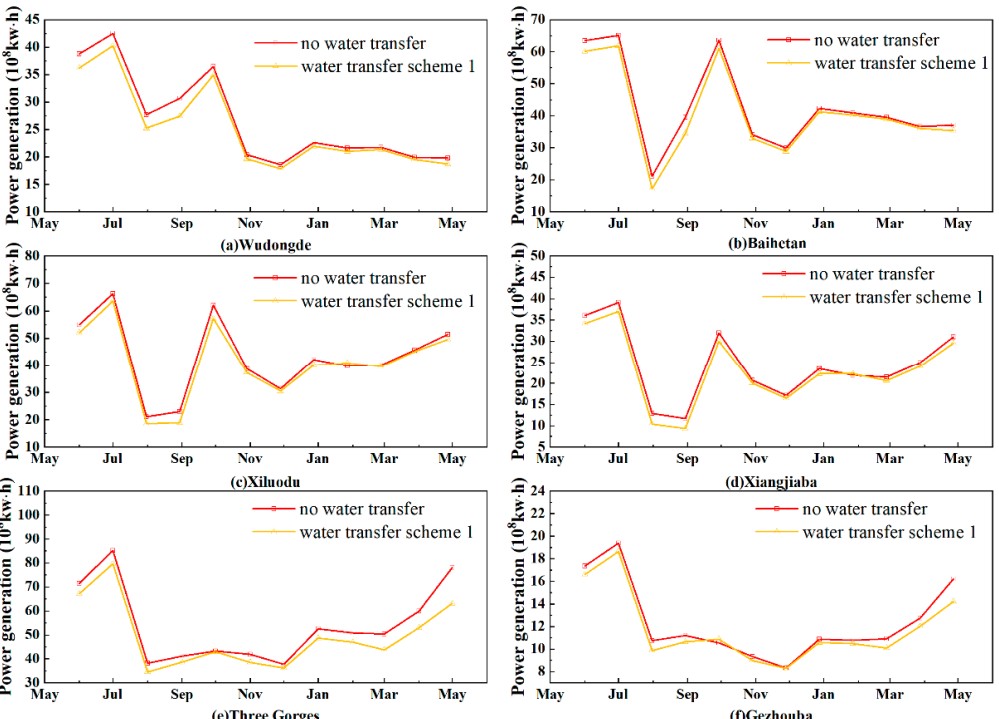

**Figure 5.** Monthly hydropower generation of cascade power stations under water diversion scheme 1 conditions.

Table 4 shows the impact of the inter-basin water transfer project on the hydropower generation of the cascade power stations under current water diversion (Scheme 1). The total hydropower generation of the cascade power stations decreased by 14.97 billion kWh, a reduction of 5.83%, compared to the situation without water diversion. Among them, the Three Gorges Power Station has the largest reduction, with a decrease of 5.79 billion kWh and a reduction rate of 8.36%. The hydropower generation of the Wudongde Reservoir, the Baihetan Reservoir, and the Xiluodu Reservoir decreased by 5.27%, 4.76%, and 4.42%.

**Table 4.** Impact of inter-basin water diversion on hydropower generation in cascade reservoirs under water diversion Scheme 1 conditions.

| Reservoir Number | | WDD | BHT | XLD | XJB | SX | GZB | Total |
|---|---|---|---|---|---|---|---|---|
| Before water diversion | wet period | 196.5 | 286.8 | 266 | 152.6 | 321.2 | 78.6 | 1301.7 |
| | dry period | 137.9 | 250.2 | 273 | 156.2 | 371.2 | 79.3 | 1267.8 |
| | Total | 334.4 | 537 | 539 | 308.8 | 692.4 | 157.9 | 2569.5 |
| After water diversion | wet period | 188.3 | 274.8 | 254.5 | 144.9 | 302.9 | 75.8 | 1241.2 |
| | dry period | 128.5 | 236.6 | 260.8 | 146.5 | 331.6 | 74.7 | 1178.7 |
| | Total | 316.8 | 511.4 | 515.3 | 291.4 | 634.5 | 150.5 | 2419.9 |
| Power loss | wet period | 8.2 | 12 | 11.6 | 7.7 | 18.3 | 2.8 | 60.6 |
| | dry period | 9.4 | 13.6 | 12.2 | 9.7 | 39.6 | 4.7 | 89.2 |
| | Total | 17.6 | 25.6 | 23.8 | 17.4 | 57.9 | 7.5 | 149.8 |
| Reduced per-centage | wet period | 4.19% | 4.18% | 4.34% | 5.07% | 5.70% | 3.52% | 27.00% |
| | dry period | 6.81% | 5.43% | 4.48% | 6.20% | 10.67% | 5.87% | 39.46% |
| | Total | 11.00% | 9.61% | 8.82% | 11.27% | 16.37% | 9.39% | 66.46% |

Table 4 demonstrates that, under Scheme 1, the inter-basin water transfer project has an impact on the power generation of the cascade reservoirs in the Yangtze River mainstem. During the flood season, the power generation loss of the cascade power stations is 6.06 billion kWh, with a reduction rate of 4.65%; during the dry season, the power generation loss is 8.91 billion kWh, with a reduction rate of 7.03%. This indicates that the inter-basin water transfer project will have a certain impact on the power generation of the cascade power stations, especially during the dry season, when the impact is more significant.

Further analysis of Table 4 reveals that the impact of inter-basin water transfer projects on different reservoirs varies. The Three Gorges Reservoir is most affected during the dry season, with a decrease in power generation of 3.96 billion kWh, or 10.67%, while the Xiluodu Reservoir is least affected during the dry season, with a decrease in power generation of 1.22 billion kWh, or 4.48%. In addition, the power generation reduction of the Wudongde, Baihetan, and Xiluodu reservoirs during the dry season decreases in order. This is because their distance from the inter-basin water transfer project upstream gradually increases, and after passing through the reservoirs' regulation and compensation, their impact on downstream reservoirs gradually decreases.

### 4.2. Impacts of Water Transfer Scheme 2 on Reservoir Generation

In order to investigate the impact of water transfer Scheme 2 on the power generation of the Yangtze River mainstream cascade hydropower stations under extreme drought conditions, this study considers the inflow of each reservoir under extreme drought conditions and applies the impact analysis model constructed to calculate the changes in power generation of each reservoir before and after water transfer, as well as its impact on the annual power generation structure.

Table 5 presents the impact of the inter-basin water transfer project on hydropower generation of the cascade hydropower stations under the second water transfer scheme, which aims to transfer part of the dry season water to the wet season while keeping the total annual water transfer amount unchanged. Based on the inflow data of each reservoir during extreme drought conditions and using the impact analysis model developed in this study, we calculate the changes in hydropower generation of each reservoir before and after water transfer, as well as the impacts on the annual hydropower generation structure.

**Table 5.** Impact of water diversion, before and after, on power generation in each reservoir under water diversion Scheme 2 conditions.

| Reservoir Number | | WDD | BHT | XLD | XJB | SX | GZB | Total |
|---|---|---|---|---|---|---|---|---|
| Before water diversion | wet period | 196.5 | 286.8 | 266 | 152.6 | 321.2 | 78.6 | 1301.7 |
| | dry period | 137.9 | 250.2 | 273 | 156.2 | 371.2 | 79.3 | 1267.8 |
| | Total | 334.4 | 537 | 539 | 308.8 | 692.4 | 157.9 | 2569.5 |
| After water diversion | wet period | 192.3 | 280.6 | 260.7 | 147.1 | 285.6 | 74.2 | 1240.5 |
| | dry period | 127.1 | 234 | 257.5 | 145.6 | 351.5 | 77.4 | 1193.1 |
| | Total | 319.4 | 514.6 | 518.2 | 292.7 | 637.1 | 151.6 | 2433.6 |
| Power loss | wet period | 4.2 | 6.2 | 5.3 | 5.5 | 35.6 | 4.4 | 61.2 |
| | dry period | 10.8 | 16.2 | 15.6 | 10.6 | 19.6 | 2 | 74.8 |
| | Total | 15 | 22.4 | 20.9 | 16.1 | 55.2 | 6.4 | 136 |
| Reduced per-centage | wet period | 2.15% | 2.15% | 2.00% | 3.58% | 11.09% | 5.65% | 26.62% |
| | dry period | 7.84% | 6.46% | 5.70% | 6.76% | 5.29% | 2.49% | 34.54% |
| | Total | 9.99% | 8.61% | 7.70% | 10.34% | 16.38% | 8.14% | 61.16% |

The reductions in power generation of the Wudongde, Baihetan, and Xiluodu hydropower stations are 4.50%, 4.16%, and 3.88%, respectively. The reduction in power generation gradually decreases from upstream to downstream, which can be attributed to the compensating effect of the reservoir storage on downstream power stations, as the upstream diverted flow passes through the reservoirs.

Table 5 shows the impact of water diversion Scheme 2 on the power generation structure of each power station. It can be seen from the table that under water diversion Scheme 2, the power generation reduction of cascade power stations in the flood season is 61.2 billion kilowatt-hours, a decrease in 4.70%; in the dry season, the power generation reduction is 74.7 billion kilowatt-hours, a decrease of 5.89%. It can be observed that the power generation loss of cascade power stations in the dry season is greater than that in the flood season, and the power generation reduction in the dry season is also greater than that in the flood season. This is because under water diversion Scheme 2, the annual water diversion volume is evenly distributed to each month while ensuring that the total annual water diversion volume in the water diversion area remains unchanged, and some of the water diverted from the flood season is transferred to the dry season.

The Three Gorges Reservoir is most affected by water diversion during the flood season, with a reduction of 35.6 billion kilowatt-hours, corresponding to a decrease of 11.09%. In contrast, the impact of water diversion on power generation in the flood season is smallest at the Xiluodu Reservoir, with a reduction of 5.3 billion kilowatt-hours, a decrease in only 2.00%. The power generation reductions for the Wudongde, Baihetan, and Xiluodu Reservoirs in the flood season are 2.15%, 2.14%, and 2.00%, respectively, decreasing gradually with increasing distance from the upstream inter-basin water diversion project and the compensatory effect of the reservoirs.

*4.3. Comparative Analysis of Water Diversion Scheme 1 and Scheme 2*

Table 6 shows a comparison of the effects of two water transfer schemes on power generation and power structure of the cascade hydropower stations in the Yangtze River during extremely dry years. The data indicate that under water transfer Scheme 1, the power generation of the cascade hydropower stations in the Yangtze River is 2419.8 billion kWh, while under water transfer Scheme 2, the power generation is 2433.6 billion kWh. Compared with water transfer Scheme 1, water transfer Scheme 2 reduces the power loss by 1.38 billion kWh, and the reduction rate of power generation decreases by 0.54%. This indicates that, under the condition of constant total water transfer, water transfer Scheme 2 can reduce the impact of inter-basin water transfer on power generation of the cascade hydropower stations in the Yangtze River. The power loss for water transfer Scheme 2 mainly occurs in the flood season, while the power loss for water transfer Scheme 1 mainly occurs in the dry season. Since the power generation during the dry season is already insufficient, water transfer Scheme 1 would aggravate the problem of insufficient power supply during the dry season of the Yangtze River mainstem.

**Table 6.** Comparative results of two water diversion schemes on the hydropower generation and electricity structure of cascade hydropower stations.

| Water Transfer Schemes | Power Generation after Water Transfer ($10^8$ KW·h) | | | Loss of Generation Compared to No Water Transfer ($10^8$ KW·h) | | | Reduction Compared to No Water Transfer | | |
|---|---|---|---|---|---|---|---|---|---|
| | Wet Period | Dry Period | Total | Wet Period | Dry Period | Total | Wet Period | Dry Period | Total |
| 1 | 1241.2 | 1178.7 | 2419.8 | 60.6 | 89.1 | 149.7 | 4.65% | 7.03% | 5.83% |
| 2 | 1240.5 | 1193.1 | 2433.6 | 61.2 | 74.7 | 135.9 | 4.70% | 5.89% | 5.29% |

Figure 6 shows that the trends in the decrease in power generation for the Wudongde, Baihetan, Xiluodu, and Xiangjiaba reservoirs are relatively consistent before and after water diversion. However, the power generation in the Three Gorges and Gezhouba reservoirs decrease significantly in April and May, due to a large proportion of water diversion during these two months, which result in reduced inflow to the upstream reservoirs and, consequently, reduced power generation in the Three Gorges and Gezhouba reservoirs.

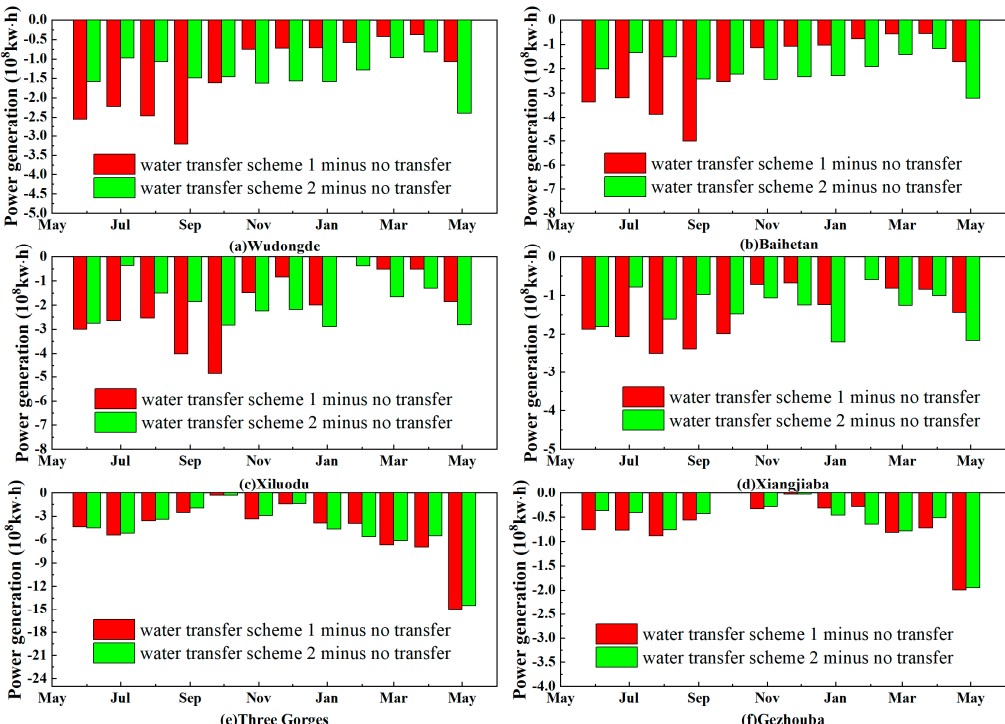

**Figure 6.** Power generation of each reservoir under two water diversion schemes.

## 5. Discussion

There is a complex interaction between water diversion projects and hydropower generation in the Yangtze River Basin [36]. In this study, it was found that the power generation of the cascade reservoirs along the main stem of the Yangtze River significantly decreased during water diversion, particularly during the dry season when the water demand in the receiving basin for inter-basin water transfer was concentrated, while the water diversion volume during the wet season was higher than that during the dry season. Analysis of the months with the maximum power generation loss for each reservoir revealed that the power generation loss was highest in April and May, when the water diversion volume was large, and the reservoir was in the period of water level decline, which further increased the power generation loss due to the reduced head.

In addition, this study found that the Three Gorges Reservoir had the largest reduction in annual power generation after water diversion. This might be due to its proximity to the Baihetan Water Diversion Project, the South-to-North Water Diversion West Route Project, and the Hanjiang River Diversion Project, whose water intake locations were relatively close to the Three Gorges Reservoir and received less inflow, leading to a direct reduction in the inflow to the Three Gorges Reservoir and a subsequent decrease in power generation. However, the percentage of power generation loss for the Wudongde, Baihetan, and Xiluodu reservoirs gradually decreased from upstream to downstream, possibly due to the gradually increasing distance between these reservoirs and the inter-basin water transfer projects upstream, which reduced the impact of inter-basin water transfer projects on downstream reservoirs. Therefore, a comprehensive assessment of the impact of water diversion projects on hydropower generation in the water intake areas is necessary for sustainable water resource utilization.

This study compared and analyzed two water diversion schemes of inter-basin water transfer projects, namely, water diversion Scheme 1 and water diversion Scheme 2. Water diversion Scheme 1 adjusted the water diversion volume on a monthly basis to achieve the optimal water diversion effect based on the design scheme and parameters in the initial report. Water diversion Scheme 2 distributed the annual water diversion volume evenly among the months under the premise of a constant total water diversion volume to

achieve balanced and stable water diversion. Water diversion Scheme 2 reduced the power generation loss by 1.38 billion kWh compared to water diversion Scheme 1. The results indicate that changing the annual distribution of water diversion could reduce the impact of water diversion on power generation of cascade hydropower stations in the water intake areas, but cannot completely offset the impact of water diversion on power generation.

Through quantitative analysis of hydropower generation and power structure of cascade hydropower stations, this study provides new suggestions for decision making regarding the water diversion scheme and the engineering demonstration of inter-basin water transfer projects. It is recommended to adopt the optimal water diversion scheme by considering factors such as power generation, water consumption in the water intake area, and reservoirs in the design of inter-basin water transfer projects. Additionally, the findings of this study can also serve as a reference for the optimization of water diversion schemes in other inter-basin water transfer projects. With the increasingly serious global water shortage, inter-basin water transfer has become an important way to solve the water scarcity problem. Therefore, scientific and rational technical support and policy guarantees are needed in the construction of inter-basin water transfer projects, which involve multiple factors, such as project investment, environmental protection, and social stability. The research results of this study provide valuable information for decision making in water diversion projects and hydropower generation fields in China and other countries, especially in the selection of the optimal water diversion method and achieving sustainable development of inter-basin water transfer projects. Thus, this study also provides important references for exploring the feasibility of solving the global water scarcity problem.

## 6. Conclusions

In order to evaluate the impact of inter-basin water transfer projects on the power generation of cascade hydropower stations in the upstream Yangtze River Basin during extreme drought years, this study established a CLHMS-LSTM model to analyze the effects of such projects on hydropower generation, and investigated the impact of two different inter-basin water transfer schemes on the cascade hydropower stations. The main conclusions of this study are:

(1) The CLHMS-LSTM model can be used to effectively characterize the hydrological characteristics of the main stem of the Yangtze River. Additionally, it can simulate the reservoir operation well.

(2) Inter-basin water transfer projects decrease hydropower generation of cascade hydropower stations in the Yangtze River.

(3) Comparing two inter-basin water transfer schemes, Scheme 2 was found to reduce the loss of hydropower generation in the Yangtze River cascade hydropower stations without decreasing the total amount of water transferred. Compared to Scheme 1, Scheme 2 could reduce the loss of hydropower generation by 1.38 billion kilowatt-hours.

(4) Comparing the impact of the two inter-basin water transfer schemes on the hydropower generation structure of each power station, both schemes resulted in decreased hydropower generation mainly during the wet season. However, Scheme 2 caused less hydropower generation loss during the dry season compared to Scheme 1, making it more beneficial for addressing the problem of inadequate power supply during the dry season and the greater pressure of hydro peak regulation.

This study examined how inter-basin water transfer projects affect hydropower generation and structure in the Yangtze River cascade hydropower stations. The study analyzed two water transfer schemes and found that inter-basin water transfer Scheme 2 could reduce the impact on hydropower generation during the dry season, compared to inter-basin water transfer Scheme 1. Future research will improve the joint operation mode of cascade hydropower stations [37], consider the effects of runoff compensation and reservoir capacity compensation, and optimize the model in order to analyze the impact of inter-basin water transfer projects on hydropower generation. This study provides new ideas for the

decision-making process of other inter-basin water transfer projects and the formulation of water transfer schemes by quantitatively analyzing the changes in annual hydropower generation and the hydropower generation structure.

**Author Contributions:** F.W. designed and conducted the experiments. F.W., M.Y., N.D. and W.G. wrote the draft of the paper. J.C. proposed the main structure of this study. Y.Z., H.W. and X.L. provided useful advice and made some corrections. All authors have read and agreed to the published version of the manuscript.

**Funding:** This research was funded by the Research Programme of the China Three Gorges Corporation "Impact of trans-basin water diversion on the Yangtze River Basin and its adaptation" (No. 0704183), the Research Programme of the Kunming Engineering Corporation Limited (No. DJ-HXGG-2021-04), the Open Research Fund of the Key Laboratory of Flood and Drought Hazard Control of the Ministry of Water Resources (KYFB202112071051), the Belt and Road Special Foundation of the State Key Laboratory of Hydrology-Water Resources and Hydraulic Engineering (2021490311),and the Key Research and Development Programme of Yunnan (No. 202203AA080010) as part of the Science and Technology Plan Project of Yunnan Provincial Department of Science and Technology.

**Institutional Review Board Statement:** Not applicable.

**Informed Consent Statement:** Not applicable.

**Data Availability Statement:** Not applicable.

**Conflicts of Interest:** The authors declare no conflict of interest.

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
