# Peer review of "The Impact of Inter-Basin Water Transfer Schemes on Hydropower Generation in the Upper Reaches of the Yangtze River during Extreme Drought Years"

_sustainability, doi:10.3390/su15108373_

Round 1
Reviewer 1 Report
Q1:In line 184, it is mentioned that "the operation models for the Wudongde and Baihetan reservoirs adopted conventional reservoir operation models based on their respective operating rules." Please briefly introduce the conventional operation models for reservoirs ?
Q2:In line 202, it was mentioned that "we collected preliminary design reports for each water diversion project", while in line 207, it was stated that "we adopt two different water diversion schemes in this study". Please confirm again how the two water diversion scenarios were obtained.
Q3:In line 184, ”the operation models for the Wudongde and Baihetan reservoirs adopted conventional reservoir operation models based on their respective operating rules.” Please explain what the operating rules are for the Wudongde and Baihetan reservoirs, and which operating rule is being used?
Q4:Why is it necessary to design water transfer scheme 2, and what is the basis for designing water transfer scheme 2?
Q5: Could you provide more details about the CLHMS-LSTM model, including its assumptions and limitation? This will help readers evaluate the validity of the model and the study's findings.
Q6: In the methods section, Could you provide more information about how the data were collected and processed? This will help readers understand the reliability of the data and the potential sources of error.
Q7:In the discussion section, Could you provide more context about the policy implications of the study, including how the findings can inform decision-making about inter-basin water transfer projects and hydropower generation in China and other countries? This will help readers understand the broader significance of the study.
Specific details
Line 87: we used a coupled -> we use a coupled
Line 96: we compared and analyzed -> we compare and analyze
Line 104:“Finally, Section 5 discusses the results.”Section 6 does not mention.
Line 141: 39/9.6/80 -> 39.0/9.6/80.0
Line 273: After the water transfer -> After water diversion
Line 276: after the water transfer -> after water diversion
Table 8: KW.h -> KW•h
Specific details
Line 87: we used a coupled -> we use a coupled
Line 96: we compared and analyzed -> we compare and analyze
Line 104:“Finally, Section 5 discusses the results.”Section 6 does not mention.
Line 141: 39/9.6/80 -> 39.0/9.6/80.0
Line 273: After the water transfer -> After water diversion
Line 276: after the water transfer -> after water diversion
Table 8: KW.h -> KW•h
Reviewer 2 Report
I feel this manuscript has addressed a very important issue of current time where water is needed for sustaining all forms of life.
1. The manuscript was very well structured and written in good organized way.
2. The manuscript included all the good statistical models for measuring the water flows and data analysis used in water measurement.
3. Tables and figures showed good variation of the water measurement in different sources.
4. Language needs bit improvement to increase understanding in scientific manner.
5. Overall the project and study has great significance in the water flow measurement with determining the characteristics of the water resources.
Your article is good just improve the language of manuscript.
Reviewer 3 Report
The manuscript entitled "The impact of inter-basin water transfer schemes on hydro-2 power generation in the upper reaches of the Yangtze River 3 during extreme drought years" by Fan Wen et al. for publication in this journal. I have gone through the manuscript and the work done by the authors is new and has scientific significance. Therefore, I recommend the publication of this manuscript in this journal after minor corrections as given below:
1. In the introduction section, the author needs to discuss some recent reports.
2. The aim of the paper should be a little more highlighted in the last paragraph of the introduction.
3. Please check the entire manuscript for grammatical errors.
4. Go through the following recent article and cite them accordingly- https://doi.org/10.3390/su11123284
5. A conclusion should be rewritten in a more concise way.
Must be improved. A lot of grammatical errors can be seen.
Reviewer 4 Report
This manuscript is in good shape. It is well organized and clearly written. I suggest minor corrections and revisions.
1. The water diversion quantities in Table 2 differ from those stated in the preceding text (l. 130-138) by a factor of ten.
2. The CLHMS model, which is at the core of the research, is inadequately referenced. The single reference [26], (line 145) relates to a study of temperatures in deep reservoirs -- very different from this paper! More references are needed to establish the credibility of the model and to connect to technical background. For example, the authors might cite Zhu et al., 2017, in Water.
3. Tables 4 and 5 might be combined to give readers the overall effect of diversion (Table 4) and the "wet" and "low" comparison (Table 5) in a single table. This might be done efficiently by switching rows and columns. That is, put each reservoir in a column (n=6) and put Before ("wet", "low", "total"), After ("wet", "low", "total"), etc., in rows (n=12). Same could be done with Tables 6 & 7.
In English it is more consistent to contrast either "wet" and "dry" or "high" and "low." In the context of drought, "wet" and "dry" might be better.
4. In Figure 6, the main point is to show the difference between Schemes 1 & 2. However, those differences are very small and hard to distinguish relative to the larger seasonal changes. The seasonal change has already been shown in Figure 5, so I think Figure 6 would be more informative if each panel showed the month by month differences of Scheme 1 and Scheme 2 relative to the “no transfer” situation. Each panel of Figure 6 would then contain just two lines which would be much further apart and easily distinguished.
Use of English is good.
